# Murine Bone Marrow Erythroid Cells Have Two Branches of Differentiation Defined by the Presence of CD45 and a Different Immune Transcriptome Than Fetal Liver Erythroid Cells

**DOI:** 10.3390/ijms242115752

**Published:** 2023-10-30

**Authors:** Olga Perik-Zavodskaia, Roman Perik-Zavodskii, Kirill Nazarov, Marina Volynets, Saleh Alrhmoun, Julia Shevchenko, Sergey Sennikov

**Affiliations:** Laboratory of Molecular Immunology, Federal State Budgetary Scientific Institution Research Institute of Fundamental and Clinical Immunology, 630099 Novosibirsk, Russia; perik.zavodskaia@gmail.com (O.P.-Z.); zavodskii.1448@gmail.com (R.P.-Z.); kirill.lacrimator@mail.ru (K.N.); mrsmarinavolynets@gmail.com (M.V.); saleh.alrhmoun1@gmail.com (S.A.);

**Keywords:** mice, erythropoiesis, erythroid cells, erythroblasts, CD71+ erythroid cells, CECs, bone marrow, fetal liver, CD44, CD45

## Abstract

Mouse erythropoiesis is a multifaceted process involving the intricate interplay of proliferation, differentiation, and maturation of erythroid cells, leading to significant changes in their transcriptomic and proteomic profiles. While the immunoregulatory role of murine erythroid cells has been recognized historically, modern investigative techniques have been sparingly applied to decipher their functions. To address this gap, our study sought to comprehensively characterize mouse erythroid cells through contemporary transcriptomic and proteomic approaches. By evaluating CD71 and Ter-119 as sorting markers for murine erythroid cells and employing bulk NanoString transcriptomics, we discerned distinctive gene expression profiles between bone marrow and fetal liver-derived erythroid cells. Additionally, leveraging flow cytometry, we assessed the surface expression of CD44, CD45, CD71, and Ter-119 on normal and phenylhydrazine-induced hemolytic anemia mouse bone marrow and splenic erythroid cells. Key findings emerged: firstly, the utilization of CD71 for cell sorting yielded comparatively impure erythroid cell populations compared to Ter-119; secondly, discernible differences in immunoregulatory molecule expression were evident between erythroid cells from mouse bone marrow and fetal liver; thirdly, two discrete branches of mouse erythropoiesis were identified based on CD45 expression: CD45-negative and CD45-positive, which had been altered differently in response to phenylhydrazine. Our deductions underscore (1) Ter-119’s superiority over CD71 as a murine erythroid cell sorting marker, (2) the potential of erythroid cells in murine antimicrobial immunity, and (3) the importance of investigating CD45-positive and CD45-negative murine erythroid cells separately and in further detail in future studies.

## 1. Introduction

Murine erythropoiesis is a complex process that involves the proliferation, differentiation, and maturation of erythroid progenitor cells. Erythroid cells are composed of proerythroblasts, early and late basophilic erythroblasts, polychromatophilic erythroblasts, and orthochromatophilic erythroblasts. Orthochromatophilic erythroblasts then lose their organelles and nuclei to become erythrocytes [1]—cells whose main function is oxygen transport [2]. 

Murine erythroid cells are known to have two main markers: Ter-119 and CD71. Ter-119 is present on all erythroid cells from the proerythroblast stage onwards [3,4] and was found to be the erythroid cells’ selection marker as it is exclusively expressed on erythroid cells [5]. CD71, a receptor for holotransferrin [6], on the other hand, is present on [7,8,9,10,11] but not exclusive to erythroid cells [12,13,14,15] and, therefore, could only be treated as an enrichment marker. CD71 is also an important maturation marker of both human and murine erythroid cells—its expression drops as erythroid cells maturate [10]. Despite it being an enrichment marker, a lot of researchers are using it (and not Ter-119) to study erythroid cells, and these CD71-enriched erythroid cells are simply called CD71 erythroid cells or CECs [8,9,11].

Other surface markers that are present on the erythroid cells are under scrutiny as well. CD44 (along with the forward scatter (FCS)) is used to determine erythroid cells’ stages of differentiation [16,17]. Another erythroid cell surface marker that has caught the attention of researchers is CD45. At first, CD45 was thought to be a negative marker of murine erythroid cells [18]. However, it was later found to be, in fact, present in murine erythroid cells. Researchers even proposed it as a marker of early-stage erythroid cells [19], while others highlighted that CD45-positive erythroid cells have more prominent immunosuppressive properties compared to CD45-negative erythroid cells [20,21,22]. 

Erythroid cells are known to have a broad range of immunoregulatory properties in the form of cytokine production [23,24,25,26]. They also have potent immunosuppressive properties in the form of TGF-beta1 production and enzymatic arginine depletion by Arginase-1 and Arginase-2 enzymes [8,9,11].

Erythropoiesis as a process could disturbed by outside factors, such as hemolysis-inducing agents. We chose phenylhydrazine (PHZ)-induced hemolytic anemia (HA) for this study as it is a convenient and widely applicable model [27,28]. PHZ’s reaction with heme results in the formation of phenyliron-heme complexes, benzene, superoxide anion, and hydrogen peroxide [29]. It has also been known that lipid peroxidation occurs within the cellular membranes of erythrocytes and that protein-protein crosslinking occurs within the supporting network associated with the membrane. Processes induced by PHZ also cause destabilization of the globin portion of hemoglobin, leading to denaturation and precipitation that lead to the lysis of hemoglobin-rich cells, such as late erythroid cells, reticulocytes and erythrocytes [30].

In this study, we decided to benchmark erythroid cells’ selection and enrichment markers—CD71 and Ter-119 via bulk immune transcriptomics by NanoString, perform bulk immune transcriptome profiling of erythroid cells from the normal bone marrow and the fetal liver by NanoString, perform bulk immune secretome profiling of normal and phenylhydrazine-treated erythroid cells from the bone marrow and the spleen liver by Bio-Plex, and investigate CD44, CD45, CD71, and Ter-119 expression on normal and phenylhydrazine-treated bone marrow and spleen erythroid cells by flow cytometry and the advanced methods that came from the field of single cell transcriptomics—dimensionality reduction and clustering.

## 2. Results

### 2.1. Usage of CD71 as a Murine Erythroid Cell Enrichment Marker for Bulk Immunomics Results in B-Cell Contamination, as Well as Artificially Lower Gene Expression of Cytokines Compared to Ter-119

In order to pick the best selection marker for erythroid cells for bulk immunomics (immune transcriptomics, immune proteomics, etc.), we performed a differential gene expression analysis between the CD71 and the Ter-119 magnetically separated erythroid cells from the murine bone marrow (Figure 1).

Genes that were up-regulated in CD71-enriched erythroid cells (in CECs) were: *Ctsg*, *Cd97*, *Irf4*, *Ccr7*, *Cd69*, *Pax5*, *Fcgr2b*, *Cd14*, *C3*, *Irak3*, *Nfkbia*, *Stat3*, *Camp*, *Irf1*, *Ceacam1*, *Ets1*, *Prkcd*, *Ifngr1*, and *Cd19*.

Genes that were up-regulated in Ter-119-selected erythroid cells were: *Psmb7*, *Psmc2*, *Itga4*, *C1qbp*, *Prim1*, *Abcb10*, *Ccrl2*, *Pml*, *Cd82*, *Tfrc*, *Cd24a*, *Icam4*, *Tnfrsf14*, *Tal1*, *C1qb*, *Cd36*, *Il1b*, *Il1r2*, and *Cxcl12*.

Several differentially expressed genes (*Cd19*, *Ccr7, Cd14*, *Irf1*, and *Irf4*) in CECs belonged to B-cells [31] and/or monocytes [32], and the differentially expressed genes in Ter-119-selected erythroid cells showed a clear erythroid signature (*Cd36*, *Itga4*, *Tal1*, and *Tfrc*) [33,34]. We considered Ter-119 to be the marker of choice for murine erythroid cell separation and subsequent bulk immunomic studies.

### 2.2. Murine Bone Marrow Erythroid Cells Have an Antimicrobial Gene Expression Signature, While Murine Fetal Liver Erythroid Cells Have an Antiviral Gene Expression Signature

We then performed an immune transcriptome study of Ter-119-selected murine erythroid cells from the adult bone marrow and the fetal liver. We found the expression of many genes in the NanoString Immune Response V1 mouse panel.

Genes with detected expression (sorted in the descending mean detected probe count order) were *S100a9*, *S100a8*, *Cd24a*, *Camp*, *Tfrc*, *B2m*, *Clu*, *Tyrobp*, *Ctnnb1*, *Cd36*, *Ctss Arhgdib*, *Itgb1*, *Psmc2*, *Lilrb4*, *Cd164*, *Il1r2*, *Cd74*, *Cybb*, *H2-Aa*, *Tal1*, *Fn1*, *Cd9*, *C3 Tnfrsf14*, *Psmb7*, *Mif*, *Xbp1*, *Mx1*, *Plaur*, *Cd81*, *Cd82*, *C1qbp*, *Tnfaip3*, *Tgfb1*, *Psmb5 Mapk1*, *Ncf4*, *Cxcr4*, *Syk*, *Jak1*, *Bcap31*, *Cd44*, *Tollip*, *Ccl3*, *Itga4*, *Rae1*, *Cfi*, *Fcer1g*, *Icam4*, *Psmd7*, *Prim1*, *Ccl9*, *Cebpb*, *Ccl2*, *Nfkbiz*, *H2-K1*, *App*, *Bax*, *Abcb10*, *Il1b*, *Cdkn1a*, *Irgm1 Ets1*, *Ccr2*, *Fcgr2b*, *Ube2l3*, *Ptprc*, *Fcgr3*, *Emr1*, *C1qb*, *Litaf*, *Ctsg*, *Trem1*, *Ifnar1*, *Mapk14*, *Cxcl12*, *H2-Ab1*, *Cd2*, *Casp3*, *Cd55*, *Pml*, *Prkcd*, *Stat3*, *Tnfrsf11a*, *Clec4e*, *H2-Ea-ps*, *Ccrl2 Ptpn6*, *Ifngr1*, *Ifna1*, *Itgam*, and *Jak2*. We did not detect any *Il10* gene expression; *Arg1* and *Arg2* genes were not included in the panel.

We then performed differential gene expression analysis between the adult bone marrow and fetal liver erythroid cells. We used a more stringent than usual fold change criterion (log2(FC) > 2/log2(FC) < −2) to avoid any gene expression differences due to the different proportions of erythroid cells at the different stages of differentiation present.

Adult bone marrow erythroid cells’ enriched genes were *Camp*, *S100a9*, *S100a8*, *H2-Ea-ps*, *Cd74*, *Cybb*, *H2-Ab1*, *Cxcl12*, *Il1r2*, *H2-Aa*, *Ctsg*, *Tnfaip3*, *Casp3*, *Arhgdib*, *Nfkbiz*, Cxcr4, *C3*, *Cd79b*, *Cd2*, *Ncf4*, *Itgam*, *Trem1*, *Ets1*, *Ptprc*, *Cd24a*, *Cd82*, and *Plaur*.

Fetal liver erythroid cells’ enriched genes were *Cdkn1a*, *Ctss*, *Itga2b*, *Clu*, *Tnfrsf11a*, *Mx1*, and *Ifna1* (Figure 2).

Genes expressed by the bone marrow murine erythroid cells were enriched in several antimicrobial immunity-related Gene Ontology Biological Process Terms (Figure 3A, Table 1), and genes expressed by the fetal liver murine erythroid cells were enriched in a “Response in Type I Interferon” Gene Ontology Biological Process Term that included *Ifna1* and *Mx1* genes. Other GO terms with high scores were not related to any fetal liver-localized process (Figure 3B).

### 2.3. Murine Bone Marrow Erythroid Cells Have Two Distinct Branches of Erythropoiesis

We performed a Hierarchical Stochastic Neighbor Embedded (HSNE) dimensionality reduction and Gaussian mean-shift (GMS) clustering of normal and post-PHZ HA bone marrow and splenic erythroid cells and found two branches of erythropoiesis: small (with low forward scatter (FSC) values) CD44^dim^ CD45-negative (CD45-negative erythroid cells for short) and large (with high forward scatter (FSC) values) SSC^hi^ CD44^hi^ CD45-positive (CD45-positive erythroid cells for short), which had a shared onset at the proerythroblast stage and included all successive stages of erythroid cells’ differentiation (Figure 4).

### 2.4. Phenylhydrazine-Induced Hemolytic Anemia Distorts Both Branches of Murine Erythropoiesis

We then studied the effects of hematopoiesis-disturbing effects on erythron by modeling hemolytic anemia (HA) using phenylhydrazine (PHZ) to study how such effects affect both CD45-positive and CD45-negative branches of murine erythropoiesis. We observed stereotypical effects of the model on CD45-negative erythroid cells—a relative decrease in late forms (orthochromatophilic erythroblasts and reticulocytes) and a relative increase in early forms (proerythroblasts and early basophilic erythroblasts) [27,28,29,30]. CD45-positive erythroid cells, however, were affected in a different way—CD45-positive erythroid cells from the bone marrow had a relative increase in CD45-positive orthochromatophilic erythroblasts (Figure 5A), while CD45-positive erythroid cells from the spleen had a relative decrease in CD45-positive polychromatophilic erythroblasts (Figure 5B).

As for the relative cellular abundance, the CD45-positive erythroid cells’ branch represented the minority cell percentage-wise (4.5–16.0%) while the CD45-negative erythroid cells’ branch represented the majority of erythroid cells cell percentage-wise (84.0–95.5%). Therefore, CD45-positive erythroid cells comprise the minor branch of erythropoiesis and CD45-negative erythroid cells comprise the major branch of erythropoiesis (Figure 5A,B).

### 2.5. Phenylhydrazine-Induced Hemolytic Anemia Causes a Decrease in Cytokine Secretion by Murine Splenic Erythroid Cells

We then studied secretomes of erythroid cells from the bone marrow and the spleen both in normal condition and after the PHZ HA. We found that erythroid cells secrete IL-1b, CCL2, and CCL3 (Figure 6A). Normal bone marrow erythroid cells secreted significantly higher IL-1b and CCL2 compared to normal splenic erythroid cells (Figure 6B). PHZ HA led to CCL3 also being secreted significantly higher by bone marrow erythroid cells compared to splenic erythroid cells (Figure 6C). PHZ HA did not alter bone marrow erythroid cell secretome (Figure 6D) and caused a slight decrease in the secretion of all detected cytokines (Figure 6E).

## 3. Discussion

In this work, we benchmarked erythroid cells’ selection and enrichment markers, CD71 and Ter-119; performed bulk immune transcriptome profiling of erythroid cells from the bone marrow and the fetal liver; and investigated CD44, CD45, CD71, and Ter-119 surface expression on bone marrow erythroid cells.

Our erythroid cell selection marker benchmarking had a clear victor—Ter-119. CECs had higher *Cd19*, *Ccr7, Cd14*, *Irf1*, *and Irf4* gene expression compared to Ter-119-selected erythroid cells. These genes are expressed in B-cells [31] and/or monocytes [32]. CECs also had lower gene expression of the erythroid cell-specific genes *Cd36*, *Itga4*, *Tfrc*, and *Tal1* [33,34] and had lower *Cxcl12* and *Il1b* cytokine gene expression compared to Ter-119-selected erythroid cells. This may be because CD71 is not exclusive to erythroid cells [12,13,14,15]. CD71 magnetic separation was initially used as an enrichment for the flow cytometry [35]—a *single-cell* method that was then subsequently used for all *bulk* immunomic methods (PCR, qPCR, and bulk RNA-seq) as well. We encourage to only use Ter-119 and not CD71 for their bulk immunomic studies of erythroid cells.

The immune transcriptome study of the Ter-119-selected erythroid cells showed the expression of several previously unmentioned genes in murine erythroid cells. We found that genes expressed by the murine bone marrow erythroid cells were enriched in several antimicrobial immunity-related Gene Ontology Biological Process Terms, which could imply pathogen-killing functions by the murine bone marrow murine erythroid cells. Supporting the latter, the top genes in the bone marrow erythroid cells (even outnumbering *Tfrc*—CD71) were *S100a8*, *S100a9*, and *Camp*. Camp and Calprotectin (dimer formed by S100a8 and S100a9) are known antimicrobial proteins [36]. This could suggest an antimicrobial role for the murine bone marrow erythroid cells. Murine bone marrow, and not fetal liver erythroid cells, also have gene expression of an MHC class II pair of molecules—*H2-Aa* and *H2-Ab1* as well *CD74* gene expression, the molecule responsible for the formation and the surface-directed transport of MHC class II molecules [37]. This set of expressed genes suggests that murine bone marrow erythroid cells are capable of antigen presentation in an MHC class II-dependent manner.

*Mx1* and *Ifna1* genes that were enriched in the fetal liver erythroid cells are known antiviral proteins [38,39]. As *Mx1* is induced by the *Ifna1* and there is an expression of the receptor for the Ifna1 (*Ifnar1* gene), we can suggest an antiviral role for the fetal liver erythroid cells as well as auto-paracrine regulation of these cells by *Ifna1*.

Our immune transcriptome study also allowed to narrow down the spectrum of the expressed cytokine genes (sorted in the descending mean detected probe count order) in both adult bone marrow erythroid cells—*Tgfb1*, *Mif*, *Ccl3*, *Il1b*, *Ccl9*, *Ccl2*, and *Cxcl12*; and fetal liver erythroid cells—*Tgfb1*, *Mif*, *Ccl3*, *Il1b*, *Ccl9*, *Ccl2*, and *Ifna1*. We found no previously described *Il10* gene expression [19]. Most of the expressed cytokines (aside from the high count tolerogenic *Tgfb1* and low detected probe count proinflammatory *Il1b* and *Ifna1*) are, in fact, chemokines, which can suggest that erythroid cells could attract other locally present immune cells towards them via chemotaxis and then induce non-specific immunosuppression in these cells via Arginase-1, Arginase-2, and TGF-beta1. As for the lack of *Il10* gene expression, we suppose that could be due to its low gene expression, as NanoString only detects transcripts with >60 copies per sample, or due to the method of erythroid cell magnetic separation (Ter-119 vs. CD71).

We also found that there are two distinct branches of murine erythropoiesis—a minor CD45-positive SCC^hi^ large erythroid cell branch and a major small CD45-negative SCC^low^ erythroid cell branch. CD45-positive erythroid cells differentiated in the same manner as the CD45-negative erythroid cells—they had gradually lost CD71 expression on their cellular surface.

We observed that our data conflicts with the hypothesis [19] that CD45 is a maturation marker in murine erythroid cells. We found that both CD45-positive and CD45-negative erythroid cells go through all known consecutive stages of erythroid differentiation.

CD44 showed similar levels of surface protein expression to CD45 in CD45-positive erythroid cells but was not restricted to CD45-positive erythroid cells—CD45-negative erythroid cells also expressed CD44 but at a lower level. As CD44 is represented mostly on CD45-positive erythroid cells, we believe that the usage of CD44 for the determination of the erythroid cells’ stages of differentiation [15,16] is unfit for the task.

Previous researchers noticed that CD45-positive erythroid cells have more prominent immunosuppressive properties compared to small erythroid cells [20,21,22]. The fact that CD45-positive erythroid cells constitute a whole separate branch of erythropoiesis allows us to consider CD45-positive erythroid cells as an immunosuppressive subpopulation of erythroid cells and that they can be selected as Ter-119, CD45-double positive cells.

As CD45-positive erythroid cells have a significantly higher measured SSC, we can assume that such cells have drastically different cytoplasmic content. The information that murine CD45-positive erythroid cells are highly immunosuppressive allows us to assume that this different cytoplasmic content could be formed up of organelles responsible for the immunosuppression.

Bone marrow CD45-positive erythroid cells also behaved differently after the phenylhydrazine-induced hemolytic anemia compared to CD45-negative erythroid cells—they saw a relative increase in CD45-positive orthochromatophilic erythroblasts and a relative decrease in CD45-negative orthochromatophilic erythroblasts. As PHZ causes HA by oxidative stress and denaturation of hemoglobin [27,28,29,30], CD45-positive erythroid cells might be more resistant to such damaging factors.

We also observed a decrease in the secretion of IL-1b, CCL2, and CCL3 by splenic erythroid cells after the PHZ HA but not by bone marrow erythroid cells. As the stage of differentiation compositions were not drastically different in the bone marrow and in the spleen after the PHZ HA, this effect might be due to the tissue of erythroid cell origin itself being a major factor in erythroid cell cytokine secretion regulation and splenic erythroid cells might be more susceptible to PHZ.

We believe that these findings could reveal new depths in erythroid cell research—both in the field of erythroid cell markers and erythroid cell immune transcriptomics.

## 4. Materials and Methods

### 4.1. Mice

We obtained mice from the vivarium of the Institute of Cytology and Genetics (Novosibirsk, Russia). Mice lived in conventional vivarium conditions with water and food access ad libitum, under the natural dark/light cycle. We crossed CBA × CBA mice to obtain fetal livers. The onset of pregnancy was recorded by the appearance of a vaginal plug in females. Euthanasia and organ harvest were performed on days 12–14 post coitum.

### 4.2. Hemolytic Anemia Model

Hemolytic anemia (HA) was modeled by the administration of phenylhydrazine (PHZ) (P26252-100G, Sigma-Aldrich, St. Louis, MO, USA). We administered PHZ in phosphate-buffered saline (PBS) intraperitoneally: the first injection was 1.2 mg of PHZ per mouse, the second injection was 0.6 mg of PHZ per mouse (after 24 h), and the third injection was 0.6 mg of PHZ per mouse (after additional 12 h). We performed organ harvesting after 4 days from the start of the experiment. Intact mice were used as controls.

### 4.3. Cell Isolation

We harvested femurs (*n* = 8 for normal bone marrow and *n* = 3 for PHZ HA), spleens (*n* = 4 for normal spleen and *n* = 3 for PHZ HA), and fetal livers (*n* = 3) from mice aseptically. We obtained bone marrow cells by marrow canal PBS washing. We obtained splenocytes and fetal liver cells by homogenizing the whole organ in a glass homogenizer. We centrifuged bone marrow, spleen, and fetal liver cells in density gradient Ficoll-Urografin (ro = 1.119 g/cm^3^) for 30 min at 322 RCF and washed them twice in PBS. 

### 4.4. Magnetic Separation

We performed magnetic separation of mononuclear cells using either anti-CD71-biotinylated antibodies (#113803, Biolegend, San Diego, CA, USA) or anti-Ter-119-biotinylated antibodies (#116203, Biolegend, San Diego, CA, USA) and streptavidin-linked magnetic beads (#480015, Biolegend, San Diego, CA, USA) according to the manufacturer’s protocols (MojoSort™ Streptavidin Nanobeads Column Protocol—Positive Selection, accessed on 30 June 2023).

### 4.5. Viability Staining

We measured either CD71-enriched or Ter-119-selected erythroid cells’ viability on a Countess 3 Automated Cell Counter (Thermo Fisher Scientific, Waltham, MA, USA) according to the manufacturer’s protocols using Trypan Blue. Trypan Blue staining showed >94% viability for the sorted erythroid cells.

### 4.6. Cell Culturing

We cultured the magnetically sorted Ter-119-positive erythroid cells in the X-VIVO 10 serum-free medium with the addition of Insulin-Transferrin for 24 h at a seeding density of 1 million per mL of the medium.

### 4.7. Harvesting the Conditioned Media of Erythroid Cells

We collected the conditioned media of erythroid cells from cells after 24 h of culturing. We performed the separation by centrifugation at 1500 rpm for 10 min; the cells’ conditioned media were then transferred into new 1.5 mL tubes with the addition of BSA up to the total concentration of 0.5% and frozen at −80 °C until the cytokine quantification.

### 4.8. Cytokine Quantification in a Culture Medium Using Bio-Plex

Fifty microliters of conditioned media from CD71-enriched erythroid cells (*n* = 5–6) were prepared for a cytokine quantification with a Bio-Plex Pro mouse cytokine 23-Plex assay (#M60009RDPD, BioRad, Hercules, CA, USA) according to the manufacturer’s recommendations and analyzed on the Bio-Plex 200 instrument.

### 4.9. Total RNA Extraction

We isolated total RNA from 500,000 either CD71-enriched or Ter-119-selected erythroid cells with the Total RNA Purification Plus Kit (Norgen Biotek, Thorold, ON, Canada). We measured the concentration and quality of the total RNA in each sample on a Qubit 4 (Thermo Fisher Scientific, USA). We froze the total RNA at −80 °C until the gene expression analysis.

### 4.10. Nanostring Gene Expression Profiling

We performed gene expression profiling with the help of the NanoString nCounter SPRINT Profiler analytical system using 100 ng of total RNA from each sample. We used nCounter Mouse Immunology v1 panel (561 immunity-related genes, 15 housekeeping genes, 6 positive [POS_A to POS_F, where POS_A is the positive control with the highest concentration and POS_F is the positive control with the lowest concentration] and 8 negative controls) to analyze the total RNA samples. The samples (*n* = 3) were subjected to a 20 h hybridization reaction at 65 °C, where 5–14 μL of total RNA was combined with 3 μL of nCounter Reporter probes, 0–7 μL of DEPC-treated water, 11 μL of hybridization buffer and with 5 μL of nCounter capture probes (total reaction volume = 33 μL). After the hybridization of the probes to targets of interest in the samples, the number of target molecules was determined on the nCounter digital analyzer. We performed normalization and QC in nSolver 4 using added synthetic positive controls and the *Gapdh*, *Rpl19*, *Ppia*, *Oaz1*, *Eef1g*, *Polr2a*, *G6pdx*, *Gusb*, *Sdha*, and *Alas1* housekeeping genes included in the panel. We then performed background thresholding on the normalized data to remove non-expressing genes. The background level was determined as the mean of the POS_F controls and the genes that were below the background level in at least one sample of the differential gene experiment were removed.

### 4.11. Differential Expression and Secretion Testing

We log2-transformed the data. We then performed differential gene expression and differential protein secretion analyses using multiple *t*-tests (we considered *q*-values < 0.01–0.001 for gene expression analyses and *q*-values < 0.05 for protein secretion analyses significant) in GraphPad Prism 9.4. The Volcano plots were created in GraphPad Prism 9.4.

### 4.12. Flow Cytometry

We washed 5 × 10^6^ cells in PBS containing 0.09% NaN_3_ and stained them with the antibodies according to the manufacturer’s protocols. We used Pacific Blue™ anti-mouse TER-119/Erythroid Cells Ab #116207, PE anti-mouse/human CD44 Ab #103024, FITC anti-mouse CD45 Ab #304006, and APC anti-mouse CD71 Ab #113820 (Biolegend, San Diego, CA, USA). We then washed the cells after 30 min of incubation in the dark with 0.5 mL PBS containing 0.09% NaN_3_. We added 7-AAD to all samples right before the cytometry. 

### 4.13. Gating of Erythroid Cells

We manually gated cells from debris, singlets from the cells, alive cells from the singlets, and, finally, Ter-119^+^ cells from the living cells Attune NxT flow cytometer software for the Attune NxT flow cytometer (Figure 1) and exported them as .fcs files. 

### 4.14. Arcsinh-Data Transformation and Data Normalization

We transformed .fcs files to .csv files using a custom Python 3 code via Jupyter Notebooks. We performed *arcsinh*-transformation [40] with the automated co-factors (co-factors were FSC = 32200.0, SSC = 32200.0, CD44 = 628.8, CD45 = 235.1, CD71 = 3.3, Ter.119 = 540.0; co-factor of 32200.0 is used for the non-fluorescent markers such as FSC and SSC to preserve their linearity) of the flow cytometry data contained in the .csv files that itself performs automated marker gating using Bartlett’ Statistics for all the markers with the R script published by Melsen et al. [41] (https://github.com/janinemelsen/Single-cell-analysis-flow-cytometry/blob/master/scripts/CSV_to_transformed_normalized_FCS_git.R, accessed on 6 July 2023). After the *arcsinh*-transformation, cells negative for the marker have their fluorescence levels converted to negative numbers or zeroes, and cells positive for the marker have their fluorescence levels converted to positive numbers (Figure 2).

We then performed simultaneous batch correction and data normalization by *fdaNorm* and exported the corrected .csv files as .fcs with the R script originally published by Melsen et al. [41] (https://github.com/janinemelsen/Single-cell-analysis-flow-cytometry/blob/master/scripts/CSV_to_transformed_normalized_FCS_git.R, accessed on 6 July 2023). In brief, *fdaNorm* registers peaks on data-representing ridge plots and then aligns the peaks thus removing any batch effect present in the data (Figure 3).

### 4.15. HSNE Dimensionality Reduction and Clustering

We normalized .fcs files into the Cytosplore app [42] and subjected them to a Hierarchical Stochastic Neighbor Embedded (HSNE) dimensionality reduction. We used FCS, SSC, CD44, CD45, and CD71 for dimensionality reduction. Stages of erythroid cells’ differentiation were defined by Gaussian mean-shift (GMS) clustering. GMS clusters were mainly defined by the expression of CD71—its expression gradually fell with each cell division and, therefore, the shift of the stage of differentiation, until it was completely absent on reticulocytes (Ter-119^+^, CD71^−^ cells) and CD45—that was present only on erythroid cells from one branch and not the other. We then exported clusters and heatmap values. Clusters were then imported into GraphPad Prism 9.4 where we calculated the percentages of cells in each cluster and performed a two-way ANOVA with Tukey correction for multiple testing (we considered *q*-values < 0.01 significant) to find statistically significant changes in the structure of erythron. Heatmap values were imported into Pandas where we performed a Z-score transformation (data standardization) and created a heatmap via *bioinfokit* (https://github.com/reneshbedre/bioinfokit, accessed on 6 July 2023).

## Data Availability

NanoString gene expression data are available at Gene Expression Omnibus with the assertion number GSE238190.

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
