# Peer review of "Murine Bone Marrow Erythroid Cells Have Two Branches of Differentiation Defined by the Presence of CD45 and a Different Immune Transcriptome Than Fetal Liver Erythroid Cells"

_ijms, 2023, doi:10.3390/ijms242115752_

Round 1
Reviewer 1 Report (Previous Reviewer 2)
Comments and Suggestions for Authors
I thank the authors for the new data presented with PHZ. The relevance of the article is much better. I hope you have nice results in SCD soon.
Author Response
We thank You for Your input towards the quality of our manuscript, potential work on SCD erythroid cells cold be very interesting!
Reviewer 2 Report (New Reviewer)
Comments and Suggestions for Authors
The message of the paper is quite interesting but there are too many imprecisions and often the explanations lack clarity. Here several points to improve or review.
Figure 1: The authors performed magnetic bead sorting of CD71+ cells and Ter119+ cells. Surprisingly, in the volcano plot, the Tfrc (Transferine receptor = CD71) gene is more enriched in the TER119 population than in the CD71 population, how is this possible? In addition, the authors indicate an erythroid signature in TER119 cells which is expected and indicate an increase in the CD44 expression, yet CD44 is not present either in the volcano plot or in the list of 19 genes overexpressed in TER119 cells, how is this possible? Finally, the threshold of the volcano plot is not acceptable, it is too low (0.87 instead of the usual 2).
Figure 2: What is the point of making GO analysis for genes found significantly overexpressed in Ter119? All the analysed genes belong to an immune transcriptome (Cf Nanostring kit).
Which housekeeping genes were used for normalization on Nsolver? SOPs are an internal control for the experiment they are not used for the normalization step. The hybridization time and the quantity of RNA used in Nanostring are also missing, thy are essential data for the use of this technology.
Figure 3: The authors show a list of genes differentially expressed in BM compared with foetal liver. It would be interesting in addition to the volcano plot perform at least a GO or GSEA analysis: which are the functions of these genes? Orthogonal validations (qPCR, WB...) would be welcome.
I recommend to combine Figures 2 and 3 into a single figure.
Figure 4: Why do the authors change and no longer compare the BM with the fetal liver? Spleen arrives suddenly without really explanations! Why PHZ treatment? what is its contribution? The CD45 cluster on the HSNe plot is intriguing; the originality of this figure is based totally on CD45 expression on mature erythroid cells, orthogonal validation is essential: for example, a FACS plot to be certain that CD45 labelling is not the result of non-specific isotypic labelling or cells could be sorted for qPCR and/or western blotting experiments.
Figure 5: A big lack of clarity: how populations are defined? By FACS gating or by HSNe? Moreover, the inter-individual variability is very important and organ composition doesn't correspond to the known data from literature for WT mice; how do the authors explain this?
For greater legibility and clarity, make rather two histograms, one for BM and one for spleen.
Figure 6: It’s very important that the authors detail that cytokine assays are performed on supernatants from cells after culture step and not directly from serum. In fact, culturing cells significantly changes cytokine composition.
Supp: Beware!!!The authors show a plot after Arcsinh transformation, on the example given in the article we have a CD71+ and CD71- population centred on the 0 axis. This type of plot is called a "butterfly plot" and corresponds to a poor transformation whose result is the creation of an artefactual population (CD71- (Log10-2)) that will interfere significantly with dimensional reduction algorithms. How is it possible to have only 49% live cells? Do maybe live cells have been eliminated from the analysis due to an incorrect Arcsinh transformation?
It’s well known that FdaNorm algorithm is based on fluorescence intensity, so it is impossible and wrong to use it to normalize FSC and SSC parameters. The authors should comment on this point.
The FDA norm was not originally published by Melsen et al but by Hahne et al.
Comments on the Quality of English LanguageThe general quality of the English is acceptable, but a big improvement is necessary concerning the general clarity of the explanations.
Author Response
Dear Review, we would like to thank You for Your points and comments!
We will try our best to explain every point of Your interest:
> Figure 1: The authors performed magnetic bead sorting of CD71+ cells and Ter119+ cells. Surprisingly, in the volcano plot, the Tfrc (Transferine receptor = CD71) gene is more enriched in the TER119 population than in the CD71 population, how is this possible?
This is expected as CD71 is one of the top expressing genes in murine erythroid cells and it is also present on other cell types, such as detected B-cells and monocytes. But, unlike Erythroid cells, B-cells and monocytes do not have high levels of CD71 expression and so they dilute the overall pool in CD71-enriched erythroid cells.
> In addition, the authors indicate an erythroid signature in TER119 cells which is expected and indicates an increase in the CD44 expression, yet CD44 is not present either in the volcano plot or in the list of 19 genes overexpressed in TER119 cells, how is this possible?
This is also expected, as murine B-cells have CD44 gene expression:
https://doi.org/10.1080/08820130500265406
https://doi.org/10.25772/QZE1-BF44
CD44 is a non-specific gene for erythroid cells and is only a part of the erythroid signature.
> Finally, the threshold of the volcano plot is not acceptable, it is too low (0.87 instead of the usual 2).
There seems to be confusion here, our threshold is 0.847 on the log2 scale which equals exactly 1.8 on the linear scale, a slightly milder than 2, but a very common threshold in the field.
> Figure 2: What is the point of making GO analysis for genes found significantly overexpressed in Ter119? All the analyzed genes belong to an immune transcriptome (Cf Nanostring kit).
This indeed made no sense. We removed it!
> Which housekeeping genes were used for normalization on Nsolver?
We used Gapdh, Rpl19, Ppia, Oaz1, Eef1g, Polr2a, G6pdx, Gusb, Sdha, and Alas1 housekeeping genes for the normalization (Figure 1). We added this information to the manuscript: "We performed normalization and QC in nSolver 4 using added synthetic positive controls and the Gapdh, Rpl19, Ppia, Oaz1, Eef1g, Polr2a, G6pdx, Gusb, Sdha, and Alas1 housekeeping genes included in the panel."
> SOPs are an internal control for the experiment they are not used for the normalization step.
As You can see in Figure 1, positive controls are indeed a part of the standard normalization procedure in nSolver 4, they serve two purposes - they correct for pipetting errors as their concentration is directly proportional to the volume of the mix pipetted to the cartridge and they are also used to remove non-expressing genes from the analysis, as NanoString cannot reliably detect genes with copies less than of POS_E control. We were even instructed by the manufacturer’s representatives that is safer to exclude such transcripts altogether to avoid false gene expression detection.
> The hybridization time and the quantity of RNA used in Nanostring are also missing, thy are essential data for the use of this technology.
We added the information on the hybridization time: «The samples (n = 3) were subjected to a 20h hybridization reaction…». As for the quality of the RNA used, we did not perform any RIN assessments as we do not currently have TapeStation or BioAnalyzer apparatus, but raw detected counts describe the RNA used as high quality as both HK and Target genes showed thousands (see Gapdh gene) and tens of thousands (see S100a9 and S100a8 genes) copies detected for the top genes (Figure 1).
FIGURE 1. nSolver data QC and normalization window of the experiment.
> Figure 3: The authors show a list of genes differentially expressed in BM compared with foetal liver. It would be interesting in addition to the volcano plot perform at least a GO or GSEA analysis: which are the functions of these genes?
That is a great idea, we added GO Biological Process analyses of the detected DEGs (Figure 2, Table 1):
Figure 2. GO BP GSEApy bubble plots.
|
Term |
Overlap |
Q-value |
Score |
Genes |
|
Positive Regulation Of Response To External Stimulus |
4/155 |
0.003 |
224.76 |
NFKBIZ, S100A9, ETS1, S100A8 |
|
Response To Lipopolysaccharide |
4/159 |
0.003 |
216.71 |
TNFAIP3, CTSG, S100A9, S100A8 |
|
Peptidyl-Cysteine Modification |
2/9 |
0.003 |
2207.74 |
S100A9, S100A8 |
|
Response To Molecule Of Bacterial Origin |
3/69 |
0.004 |
344.24 |
TNFAIP3, S100A9, S100A8 |
|
Neutrophil Chemotaxis |
3/70 |
0.004 |
337.49 |
S100A9, TREM1, S100A8 |
|
Granulocyte Activation |
2/12 |
0.004 |
1448.78 |
CTSG, CAMP |
|
Granulocyte Chemotaxis |
3/73 |
0.004 |
318.53 |
S100A9, TREM1, S100A8 |
|
Defense Response To Gram-negative Bacterium |
3/75 |
0.004 |
306.87 |
CTSG, TREM1, CAMP |
|
Neutrophil Migration |
3/77 |
0.004 |
295.92 |
S100A9, TREM1, S100A8 |
|
Defense Response To Bacterium |
4/204 |
0.004 |
150.99 |
CTSG, S100A9, S100A8, CAMP |
|
Positive Regulation Of Immune Response |
3/80 |
0.004 |
280.68 |
CD74, PTPRC, ITGAM |
Table 1. Gene set enrichment analysis of the genes up-regulated in the bone marrow erythroid cells
> Orthogonal validations (qPCR, WB...) would be welcome.
We plan to perform WB in the future, as we are quite interested in the field of erythroid cell proteomics, but, sadly, we currently do not have an opportunity to do so.
> I recommend to combine Figures 2 and 3 into a single figure.
We tried this, but with the new bulky GO BP plots Figure 2 got lost, so we would prefer for the figures to be separate!
> Figure 4: Why do the authors change and no longer compare the BM with the fetal liver?
Sadly, our fetal liver proteomics data was of poor quality and had to be removed.
> Spleen arrives suddenly without really explanations! Why PHZ treatment? what is its contribution?
We were Kindly asked by one of the Reviewers to add PHZ specifically in the spleen as he was interested in how would hemopoiesis-disturbing agents affect the CD45-positive branch of murine erythropoiesis. His suggestion was interesting, as the CD45-positive branch of murine erythropoiesis reacted differently compared to the CD45-negative branch of murine erythropoiesis.
> The CD45 cluster on the HSNe plot is intriguing; the originality of this figure is based totally on CD45 expression on mature erythroid cells, orthogonal validation is essential: for example, a FACS plot to be certain that CD45 labeling is not the result of non-specific isotypic labeling or cells could be sorted for qPCR and/or western blotting experiments.
This sole fact of CD45 expression on murine erythroid cells is not new at all.
If we just look at https://doi.org/10.1038/s42003-021-02914-4 Fig. 1: Anemia induces CECs expansion in the spleen
we would see that is now a common practice to separate murine erythroid cells into CD45+ and CD45-. Authors even observed the same thing as we:CD45+ erythroid cells had high CD44 and FSC expression. Data reproducibility is amazing.
The novelty of our work is that CD45+ erythroid cells are not a mere subpopulation, but a whole another branch of murine erythropoiesis, as CD45+ erythroid cells gradually lose CD71 expression on their cell surface - such events happen when erythroid cells divide and shift a stage of differentiation, and with their division, CD71 is diluted.
> Figure 5: A big lack of clarity: how populations are defined? By FACS gating or by HSNe?
We defined this in the materials and methods section, we used HSNE and Gaussian Mean Shift Clustering for the dimensionality reduction and clustering respectively.
We also expanded the section devoted to cluster identification towards more clarity:
«GMS clusters were mainly defined by the expression of CD71 – its expression gradually fell with each cell division and therefore – the shift of the stage of differentiation, until it was completely absent on reticulocytes (Ter-119+, CD71– cells) and CD45, that was present only on erythroid cells from one branch and not the other. »
> Moreover, the inter-individual variability is very important and organ composition doesn't correspond to the known data from the literature for WT mice; how do the authors explain this?
Murine erythropoiesis exists in e very vast range and there is no good consensus to what is normal, as erythropoiesis has been understudied due to the low interest in the topic. Murine erythropoiesis also differs from line to line and with age. We consciously collected a lot of bone marrow samples to account for this variability - just look at the expression of CD71 on Scheme 3, some samples have 0 reticulocytes (Ter-119+, CD71– cells), and some have up to 15 percent.
> For greater legibility and clarity, make rather two histograms, one for BM and one for spleen.
Done, bar plots indeed look better when separated by the organ of origin!
> Figure 6: It’s very important that the authors detail that cytokine assays are performed on supernatants from cells after culture step and not directly from serum.
We described this in detail in the materials and methods section spanning sections 4.2 – 4.7
> In fact, culturing cells significantly changes cytokine composition.
To account for this, we used a serum-free medium that does not trigger cells as much, and, as our detected cytokines are found as mRNAs in our NanoString assay, it is safe to say that they were secreted due to the initial mRNA content of the staid erythroid cells.
> Supp: Beware!!!The authors show a plot after Arcsinh transformation, on the example given in the article we have a CD71+ and CD71- population centred on the 0 axis. This type of plot is called a "butterfly plot" and corresponds to a poor transformation whose result is the creation of an artefactual population (CD71- (Log10-2)) that will interfere significantly with dimensional reduction algorithms. How is it possible to have only 49% live cells? Do maybe live cells have been eliminated from the analysis due to an incorrect Arcsinh transformation?
There seems to be confusion, we first gated erythroid cells (non-debris -> alive cells -> singlets -> Ter-119+ cells) and only after that performed variance-stabilizing arcsinh data transformation. The (CD71- (Log10-2)) population is not an artefact, these are naturally-occurring reticulocytes:
Modern flow cytometers collect data in negative values as well as positive ones, this feature is extremely helpful, for example, in the CD4/CD8 distinguishing. We just released what is hidden by default for CD71-negative reticulocytes.
Only 50% of cells alive in that exact sample were probably due to slightly high temperatures in the room of cell handling.
> It’s well known that FdaNorm algorithm is based on fluorescence intensity, so it is impossible and wrong to use it to normalize FSC and SSC parameters. The authors should comment on this point.
fdaNorm, in itself, is a variance stabilizing transformation, that also performs automated gating using a co-factor, that corresponds to the end of the linear phase of the signal increase and the beginning of the log phase of the signal increase. We both empirically and using Bartlett’ Statistics found out that supplying FSC and SSC with a high enough co-factor (32200.0 for our Attune NxT apparatus) would result in the log phase of the signal never beginning and data linearity preservation! This «linearizing» co-factor puts FSC and SSC on the same scale as other parameters (-5 to 10 interval for all markers), which is extremely important for the dimensionality reduction in Cytosplore and must be determined for every flow cytometer individually.
> The FDA norm was not originally published by Melsen et al but by Hahne et al.
We added the citation, thank You!
Overall, we would like to Thank You for the enrichment of our manuscript!

Reviewer 3 Report (New Reviewer)
Comments and Suggestions for Authors
Perik-Zavodskaia et al. present a comprehensive study on erythroid cells focusing on the two branches of differentiation defined by CD45 presence. They also discuss an important immune biology aspect of erythroid cells, which was often overlooked, precluding erythrocytes' role to transport gas. "Neutrophil Action" underlining the antimicrobial role of erythroid cells in murine bone marrow is convincing. Since erythroid cells are normally considered CD45 negative after linage restriction, the story of CD45 positive erythroid cells is interesting.
There is, however, one issue with finding cell selection markers. The authors discuss Ter119's victory over CD71 for erythrocyte isolation. This was not a surprise since CD71 declines during red blood cell maturation and terminally mature erythrocytes lack CD71, so the cells isolated with CD71 are immature, they are reticulocytes. It seems obvious that they show a different gene expression pattern and less erythroid specific genes. People using CD71 for cell selection should be well aware that they are purifying reticulocytes rather than erythrocytes. But since the victory of Ter119 was stated at the beginning and the main part of the study was done with Ter119 isolated cells, this might not have interfered with the results throughout the study. Would it be possible for the authors to comment on that? All together, the study is nicely done using state of the art methods, like BioPlex analysis and Nanostring gene expression profiling. They also performed sophisticated data analysis and presented their findings in a well-organized manner.
Minor points
Fig.6 B-E: The orange dots are labelled but he black dots are not. Please think about labelling both.
Line 213: The word “liver” seems to be missing in the sentence “…, and the fetal liver and investigated CD44, CD45, CD71…” Please reconsider the sentence.
Line 326: The sentence seems to be mixed up “We were the magnetically sorted cells cultured…” Please re-write the sentence.
Question of interest: Why did the authors add BSA to the supernatant before freezing, wouldn’t it be stable at -80°C without any supplement?
Author Response
We would like to thank You for Your input towards our manuscript. Indeed, people using CD71 for erythroid cell enrichment should be aware of the potential impurity of the method, but there were so many CD71 erythroid cells or CECs-related articles we had to test for it for fact.
> Fig.6 B-E: The orange dots are labelled but he black dots are not. Please think about labelling both.
Points (genes or Proteins) on Volcano Plot are usually labelled when they are differentially expressed or secreted and there are only 3 proteins present so we would like to keep plot as is.
> Line 213: The word “liver” seems to be missing in the sentence “…, and the fetal liver and investigated CD44, CD45, CD71…” Please reconsider the sentence.
> Line 326: The sentence seems to be mixed up “We were the magnetically sorted cells cultured…” Please re-write the sentence.
Thank You, we fixed it!
> Question of interest: Why did the authors add BSA to the supernatant before freezing, wouldn’t it be stable at -80°C without any supplement?
We used serum-free culture medium and BioRad recommend to add BSA to such media to prevent cytokine adhesion to the plastics used for storage.
This manuscript is a resubmission of an earlier submission. The following is a list of the peer review reports and author responses from that submission.
Round 1
Reviewer 1 Report
Comments and Suggestions for Authors
In the present manuscript, the authors first compare two different methods for isolation of mouse bone marrow erythrocytes, more “conventionally” and broadly used CD71 and Ter-119. Using a transcriptomic approach, they find that CD71-isolated cells contain transcripts that clearly belong to monocytes or B cells, and thus propose Ter-119-mediated isolation as a better option. They further compare gene transcription signature of Ter-119-isolated bone marrow erythroid cells and fetal liver erythroid cells. Then they perform HSNE dimensionality reduction and clustering of bone marrow cells, and conclude that 2 different branches of erythropoiesis can be found, divided into small cd45-negative cells and larger CD45-positive cells, which are less represented (less than 10%), and these two could have different physiological roles.
The findings of the study are interesting and a valuable contribution to scientists working in the field, the authors also precisely studied the outcome of transcriptomics assays and offer valuable explanations of the role of the discovered active genes. The article would however profit from a better organization, for example, designations of cell groups are listed with abbreviations and their meaning as a flowing text, and a Table would be much more convenient for the reader, especially because the repetitions within the article could be avoided. As data analysis with modern methods is an important component of the article, the criteria of categorization and data evaluation should be explained in more detail in the Material and Methods section (please also see remarks below). Several abbreviations are not explained (like HSNE, FACS-specific terminology). The manuscript (especially Materials and Methods part) should be re-read carefully and corrected for missing words, SI units, etc.
Please find below a list of remarks which I hope you will find helpful.
Line 15: and studied
Line 17: probably you mean: utilizing CD71 as a positive selection marker introduces contamination? Please reword.
Line 19: “ …murine erythropoiesis, CD45-negative…“ Comma missing
Line 20: “CD71 is not fit for bulk erythroid cell immunomics” – this sentence is vague, maybe: CD71 is not the optimal marker characteristic for all erythroid cells, or similar
Line 43: „such as CD71 enriched erythroid cells”, maybe “therefore CD71-enriched erythroid cells…”
Line 46: “CD44 (along with the FCS)”: the abbreviation FCS is not explained – please clarify
Line 50: minus sign can be interpreted as hyphenation, please stick to CD45-positive/CD45-negative spelling
Line 63: results in
Line 63: artificially lower gene expression xxx
Line 76: as several differentially expressed genes…
Line 79: “for bulk erythroid cell immunomics”: for bulk separation of erytrhroid cells for immunomics studies
Line 83: as the figure should be self-explaining, please add the criterion of exclusion to figure legend
Line 91 (and throughout the text): these are gene transcripts and not genes
Line 91: “descending mean detected probe count order”: I am very sorry, but I do not understand why the normalized detected probe count does not agree with the order on the list of genes – would you consider harmonizing the list with the Figure?
Line 98: Figure 2b is cited in the text before 2a
Line 165: “then inconspicuously translated into it being”: awkward expression; and was subsequently used for all bulk immunomic methods, or similar
Line 190: “lure” may not be a great expression here, as it literally means to trick someone to a certain place or trick them out to do something they should not do, i.e. has a negative connotation
Line 232: please use decimal point instead of comma and 3 in cubic cm should be in superscript.
Line 265: please improve on the explanation – form the text it is not clear that POS_F is the least concentrated positive control and considered detection limit.
Line 272: 5x10e6 cells
Line 272: NaN3, 3 in subscript
Line 276: after 30, time unit missing
Line 280: singlets form cell aggregates
Scheme1. Gates are drawn with a very light colour, please improve the resolution. FSC-A to FSC-H plot: what was the rationale for setting the gate (maybe better discernible in a histogram)? Viability plot is not a dot plot as the others, and the y-axis label states “percent of max”-what precisely is this, % counted cells? Why is it not possible to stick to conventional FSC against 7-AAD plot, and why is the live/dead discrimination not sharp? Is there an explanation for the two populations of CD71-positive cells within the Ter-119 positive gated cells?
Line 294: After the arcsinh-transformation, cells negative for the marker… (comma missing)
Author Response
Dear Reviewer,
We would like to thank You for your wonderful suggestions and for pointing out the errors.
We tried to correct all the shortcomings of our article clearly indicated by You and added the missing explanations of all specific abbreviations.
As for scheme 1.
>The gate is drawn in a very light color, improve the resolution.
We changed the colors and improved the resolution of the graphs.
> FSC-A to FSC-H plot: what was the rationale for setting the gate (maybe better discernible in a histogram)?
This graph is commonly used in the field to separate single cells from cell aggregates.
> Viability plot is not a dot plot as the others, and the y-axis label states “percent of max”-what precisely is this, % counted cells?
The percentage of maximum value in the Attune NxT flow cytometer's software refers to the maximum expression found overall in the sample. It's hardcoded that way.
> Why is it not possible to stick to conventional FSC against 7-AAD plot, and why is the live/dead discrimination not sharp?
We were not always able to select live singlets using FSC versus 7-AAD because sometimes there were intermediate populations, so we stuck with the 7-AAD histogram as it was way more clear. We also fixed the histogram plot, so that is correct for the sample and has a clear separation point..
> Is there an explanation for the two populations of CD71-positive cells within the Ter-119 positive gated cells?
Actually, yes, this is sample 8, the left cluster consists of reticulocytes - they are CD71-negative, the intermediate cluster consists of orthochromatophilic erythroblasts - they have the largest number of cells, and the right cluster consists of all other stages of differentiation.
Reviewer 2 Report
Comments and Suggestions for Authors
Olga Perik-Zevodskaia and colleagues describe the transcriptional differences in erythroid cells. The search for the best marker in erythropoiesis has been extensively studied for many years. The new omics allow a better characterization of cell populations. Thus, the study has an interest in the scientific community. However, the results are not very ground-breaking. It is already well-established that Ter119 is a better surface marker for murine erythropoiesis. CD71 is known as a marker shared with other hematopoietic cells, although its surface levels decrease during erythropoiesis and are absent in mature red cells. The second part of the manuscript, comparing the bone marrow and the fetal liver erythroid cells and the differences based on CD45, is more relevant.
Before publication, there are some specific concerns to address:
1- The abstract needs to be extensively improved. For example, the work's aim is not clearly indicated, and expressions like “we believe that” should be removed.
2 2 - Please, discuss what’s happening in human erythroid cells. The use of GlyA as an erythroid marker is well described, but please show information for CD45.
3-The manuscript's impact will be enhanced by showing data on stress or disordered erythropoiesis. I suggest performing an experiment with mice treated with phenylhydrazine and monitoring the frequency of erythroid cells based on CD45 expression in the spleen. I would also include information on erythroid disorders like sickle cell disease.
4 4- These articles should be referenced: 10.1016/j.immuni.2018.08.019; 10.1111/bjh.15902
Comments on the Quality of English LanguageSome expressions as “we believe that” (abstrcat( or "was a clear winner" (line 79) should be changed.
Author Response
Dear Reviewer,
We would like to thank You for Your wonderful suggestions and for pointing out the errors.
We tried to correct all the shortcomings of our manuscript, indicated by You.
0 - While it might be clear for most researchers that Ter-119 is the marker of choice, allmost every recent immunological paper describes the use of CD71 as the erythroid cell sorting marker: https://www.frontiersin.org/articles/10.3389/fimmu.2020.597433/full
https://www.frontiersin.org/articles/10.3389/fimmu.2021.705197/full
https://www.frontiersin.org/articles/10.3389/fimmu.2023.1131379/full
https://www.frontiersin.org/articles/10.3389/fimmu.2022.830025/full
1- The abstract needs to be extensively improved. For example, the work's aim is not clearly indicated, and expressions like “we believe that” should be removed.
We have completely rewritten the abstract to meet the high standards of the IJMS journal.
2 - Please, discuss what's happening in human erythroid cells. The use of GlyA as an erythroid marker is well described, but please show information for CD45.
We are currently in the process of writing a large paper on human erythroid cells and would be happy to discuss the topic of selection marker there. We also thank you for the link to the article on sickle cell anemia in humans (https://onlinelibrary.wiley.com/doi/10.1111/bjh.15902), we will be happy to quote it in our future manuscript!
3 -The manuscript's impact will be enhanced by showing data on stress or disordered erythropoiesis. I suggest performing an experiment with mice treated with phenylhydrazine and monitoring the frequency of erythroid cells based on CD45 expression in the spleen. I would also include information on erythroid disorders like sickle cell disease.
My Colleague and co-author of this article, Kirill Nazarov, is also currently writing a large article on the effect of phenylhydrazine-induced anemia, acute hypoxia and acute blood loss on both branches of erythropoiesis and their secretion, and this article in our vision is the starting point for indicating the presence of two branches mouse erythropoiesis.
4 Thank you for pointing out the important article on CD45-positive erythropoiesis progenitors (https://pubmed.ncbi.nlm.nih.gov/30314756/), we have added a reference in the text!
Round 2
Reviewer 2 Report
Comments and Suggestions for Authors
I thank the authors fro addressing all the points. I still consider the abstract needs to be improved to attract the reader
Comments on the Quality of English LanguageThe abstract is much better than in the first version, but still they need to improve it.
Author Response
Dear Reviewer, we remade the annotation, we hope it is better now!
Round 3
Reviewer 2 Report
Comments and Suggestions for Authors
Thanks for the revised of your manuscript. However, after my first revision, my recommendation to the editor was rejecting you manuscript. The manuscript has not the novelty enough for publication without some of the experiments proposed. I encourage including new data along with the one shown here.